# Teaching Machine Learning in the Context of Critical Quantitative Information Literacy

**Carrie Diaz Eaton** [1]

## Abstract

Bates College, is a small liberal arts postsecondary institution in the northeast United States. An information literacy course, Calling Bull, serves as an introductory data science class as well as a prerequisite-free quantitative literacy class. In this context, we spend a week discussing machine learning, with an emphasis on facial recognition algorithms. The emphasis is on the general algorithmic approach, critical inquiry of the process and careful interpretation of results presented in research or decision-making. This module relies on the use of open educational materials, discussion, and careful attention to issues of marginalization and algorithmic justice.

## 1. Introduction

Calling Bull is an open educational course developed by quantitative biologist Carl Bergstrom and information scientist Jevin West. Their website

**http://callingbullshit.org/**

(and the secondary-level friendly companion website callingbull.org), has a suggested syllabus, case studies and links to lectures on YouTube (Bergstrom & West, 2021). Bergstrom and West published a book by the same name in 2020.

The goal of the Calling Bull course they developed is to prepare students for a world of big data by introducing students to tools and techniques for sorting through information in the data economy. The curriculum also helps students understand what drives information in the digital and scientific ecosystems.

In 2019, I adopted and adapted Calling Bull for a specific context at Bates College in the Digital and Computational Studies program as part of a broader data science set of courses. Bates College is a small liberal arts school in the United States. The Digital and Computational Studies program is a new interdisciplinary department designed to bridge multiple disciplines with computer science, computation, and exploration of the digital ecosystem.

The goals of the Calling Bull course from an institutional perspective were to 1) collaborate with a theme of interest to social science majors, 2) provide a gentle introduction to programming with R, 3) reinforce key skills such as interpreting graphs and understanding uncertainty in science, and 4) meet the quantitative literacy standards. Eventually, the course became a gentle introduction to critical data science and an introduction to the digital and computational studies program.

At most postsecondary institutions in the United States, all students are required to take one a quantitative literacy course, regardless of their major. Many times their department offers such a course, but if not other departments, such as mathematics offer an option. I found Calling Bull a particularly compelling theme for a quantitative literacy course because it captures key quantitative reasoning and critical thinking skills that all graduates should have. This is important for civic engagement, but also relevant in the current infodemic on the heels of the pandemic.

While the process of science is discussed and practiced in this course, the emphasis is primarily on critique and "calling bull" on misuse and miscommunication of data-based information. Our current students came of age in the big data economy, already well versed in and using the affordances of technology that the data economy has produced. However, they do not yet have the tools or practice thinking deeply, so that they can critique what the big data economy produces when necessary.

In the syllabus for the course is the following explanation of learning objectives[1]:

> **This course is designed as a community learning journey.** Together, we will:

---

[1]Digital and Computational Studies, Bates College, Lewiston, Maine, USA. Correspondence to: Carrie Diaz Eaton <cdeaton@bates.edu>.

*Proceedings of the $2^{nd}$ Teaching in Machine Learning Workshop*, PMLR, 2021.

[1]Quotes are borrowed with permission directly from the syllabus on the Calling Bull website (Bergstrom & West, 2021)

- Metacognitively engage in contemporary issues in equity and social justice related to their digital world, community, and identity. *"Think about this class every time you hear the news, make daily choices, or even put your shoes on."*
- Play with computational ideas creatively, using a growth mindset which values revision and experimentation and demonstrate community leadership skills as a collaborator that shares strengths, builds weaknesses, and contributes to a broader shared understanding. Participate in teamwork in respectful ways that allow people to relax and play with ideas.
- Recognize and translate between algebraic, numeric, visual, and verbal representations of data. *"Remain vigilant for bull contaminating your information diet and recognize said bull whenever and wherever you encounter it."*
- Design models of and computationally investigate ideas in practical and professional spaces through and communicate the process and meaning to others. *"Figure out for yourself precisely why a particular bit of bull is bull, provide a statistician or fellow scientist with a technical explanation of why a claim is bull - using R and employing proper data visualization techniques where necessary, and provide your "casually racist uncle" with an accessible and persuasive explanation of why a claim is bull."*

To support these course learning objectives, I blended the Calling Bull curriculum from Bergstrom and West with programming instruction and projects in R, and supplemented with daily data visualization activities, metacognitive reflections, and other activities. An aggregated list of this and other related community contributed curriculum can be found at the Calling Bull Instructors group at `callingbull.qubeshub.org`.

An emphasis of all courses taught by Digital and Computational Studies program faculty is the explicit attention to human rights and social justice, interrogating racism, sexism, bigotry, and other forms of exclusion in the design of digital spaces and algorithms. In this context, students have multiple opportunities to ask questions about how our assumptions and biases affect our data collection, models, visualizations, interpretations and resulting communication.

Below, I present one of the lesson plans constructed for my version of the Calling Bull class. I use the Jevin and West lecture videos and suggested readings as a foundation.

These are supplemented with with other freely available resources - videos by Joy Buolamwini and a research paper by Garg et al. utilizing word embeddings (Garg et al., 2018). These supplements develop a broader view of computing ethics and human rights beyond concerns driven by phrenology pseudoscience. They also serve to broaden the voices of those represented in the conversations about the ethics, process, uses and abuses of machine learning models.

This lesson occurs about halfway through the semester, just before their mid-term independent project. A this point they have discussed definitions of bull, quick techniques for spotting bull, and common traps around interpreting data such as conditional probabilities and correlation versus causation. They have also practiced implementing basic skills in R from using it as a basic calculator up to writing scripts for reproducible research. In the prior week, students completed a linear regression project with Pearson correlation coefficient interpretation.

## 2. Machine Learning Lesson Plan

The course was originally designed as a series of one week "modules," which include lesson plans for two 80-minute classes. However, I am presenting the most recent version which was conducted online in a compressed 7 week version due to COVID-related instructional changes. All one week "modules" were reallocated to a single hour and 45 minutes synchronous session with additional asynchronous interaction requirements. In a typical two classes per week format, I would assign reading for the first day to introduce the topic and we would discuss the topic. Then the next class, we would have an R-based programming lab in the same theme, typically based on a case study. This lab would be due at the end of the week along with a written reflection. The big data module is an exception to this general rule - the case studies are part of the thematic introduction, and the "lab" is in essence a racial equity-minded intervention.

### 2.1. Pre-class assignment

Students are asked to watch or read the following items before class. With the exception of the research paper, all are from the Calling Bull website.

- Videos: Lecture 5 Big Data on YouTube

- Case Study: Criminal Machine Learning

- Case Study: Machine Learning about Sexual Orientation?

- Research Paper: Word embeddings quantify 100 years of gender and ethnic stereotypes (Garg et al., 2018).

Each YouTube "lecture" is broken into a handful of shorter

videos, and in total are typically less than one hour. During the Big Data Lecture 5 videos, Bergstrom and West introduce the basic idea of machine learning and how training models based on biased data and assumptions can lead to faulty results. They then verbally debunk the validity of two papers which use machine learning algorithms on photographs to determine criminality and sexuality. These arguments are also presented as case studies on the Calling Bull website - and students are asked to read and comment on these and the Garg et. al research paper through the use of Perusall[2] (Perusall, 2021). The use of Perusall makes it possible for students to interact around ideas in the text before class. I am able to read these responses before class and then make adaptive changes to the in-class lesson plan described below based on any misconceptions or student questions.

The research paper uses word embeddings, which combine natural language processing and machine learning methods, to explore and expose stereotype bias as expressed through language over time (Garg et al., 2018). Unlike the questionable papers highlighted in the case studies, Garg et al. showcase cutting edge interdisciplinary techniques, utilize critical quantitative inquiry, and expose bias of machine learning models, all while exploring an interesting research question. This paper provides a disciplinary bridge between computer science and quantitative critical social science in which the use of machine learning as a research tool acts against racism and sexism instead of propagating it. The use of word embeddings to measure stereotype bias leverages the fact that machine learning is biased by the data (here historical documents) with which is it represented. This expands the scope of the discussion from image classification for criminality and sexuality to using machine learning to find language associations that reveal historical perspectives on gender, race and ethnicity. Finally, the visualizations presented in the results section are accessible to the audience.

## 2.2. In-class discussion of readings

On the whole, the class uses critical pedagogy which seeks reframe the students as knowledge holders and constructors (Freire, 2000). After a data visualization activity and sharing reflections from students on their learning from the previous week, we open with a discussion of the videos and readings for the day. I begin with crowdsourcing definitions for big data and data science, after narrowing in on a co-constructed definition, I then ask "What does it mean that data science

is changing our methodologies?" We then discuss various approaches to doing science, such as hypothesis driven, data-driven, and/or mathematical model-driven using the Rule-of-Five framework (Diaz Eaton et al., 2019). This helps us derive a common understanding of language and approaches to research questions. The goal is to help students see an inclusive framework for modeling by which data science is one approach, decentering a particular idealization of any one approach in an effort to decolonize our discussion of methods.

We then break into groups[3] with the following prompts:

1. List 5 major insights from videos

2. How did these insights about big data manifest in the readings?

3. Pick at least one of the 3 readings and post on the Jamboard[4] (Google, 2021): Which reading your group picked and two ways the big data concerns or potential bullshit issues manifest in that reading.

When discussion time is up, we also review the case study findings and in broad strokes review how machine learning works in the context of image recognition[5]. Then groups would offer a group-by-group report out as we co-construct a list of potential issues presented in the case studies and the lectures more broadly. I then tie up any specific points that have not yet been mentioned in any of the case studies.

Most students understand that the phrenology studies presented in the Calling Bull Case Studies are "bullshit," because there is no reasonable biological basis for criminality or sexuality. They can also point to the bias in the sourcing of the images in the criminality case study as a source of bias. Students also typically arrive at the idea that sexuality is not a binary, which presents an issue for classification model output. The same can be asked about whether criminality should be subject to binary classification as well. Students also clearly can understand the impact and potential legal issues surrounding the use of machine learning models and image classification to detecting criminality. However, they are often unprepared when I ask "Why would someone want to classify someone's sexuality based on their face?"

---

[2]Perusall is a collaborative annotation software. The Perusall grading algorithm itself is presented in the first day of the semester as an example of a black-box proprietary algorithm driven by machine learning, and so is brought up as a reprise in this class as well. Given the critique we make in class, I relax many of conditions of the Perusall automated grading algorithm and all scores less than 100% are manually re-graded by a teaching assistant.

[3]I take care to construct the groups for this modules because of the sensitive nature of the content and discussions. When we are face-to-face, I let them choose groups, and online, I tend to group students based on what I can discern using weekly reflections about their personal challenges, struggles, and understanding about social justice in computer science.

[4]Jamboard is a virtual whiteboard, whereas an in-person group would use the classroom whiteboard.

[5]By review, I mean that a broad sketch of how machine learning for image recognition works is presented in both the videos as well as the case studies. So I bring these up to review and answer questions.

To help prompt this discussion - because, often, uncomfortable silence follows - I next ask about places in the country or world in which having a sexual identity other than heterosexual or acting on ones sexual identity is considered illegal. As of May 2021, there are 69 countries in which homosexuality has been criminalized (BBC News Reality Check Team, 2021). Fifteen states in the United States do not offer full protections against discrimination based on sexuality (Wikipedia, 2021). This allows students to consider the implications of such technology, which re-opens the door to the criminality discussion, as we ask if there are avenues of scientific inquiry that should not be pursued for the sake of scientific curiosity, particularly when the evidence for such pursuits is non-existent.

By discussing the research paper (Garg et al., 2018), students readily make the connection between research findings reported and what students know about the history of gender and racial discrimination and bias in the United States. With attention to students' questions in the Perusall assignment, we also review one or two of the findings and associated visualizations together. Discussion of this paper prepares us for the second half of the lesson plan, which emphasizes an intersectional lens on gender and racial bias in training sets and models trained on those sets.

### 2.3. Through the lens of Joy Buolamwini

To launch the second phase of the class, I ask the following two questions:

- Where do you see examples of big data being used to benefit your life?
- Where do you see examples of data or algorithms that are biased?

This conversations adds to an overall uses and abuses conversation, but with an emphasis on personal impact. Depending on the student's axes of privilege or marginalization, they may have more positive or negative personal experiences with some of the automated decision-making models that run their life, from Netflix algorithms and Google ads to credit approval and policing. This portion of the class is intended to provide insight as to the continued invisibility and disregard for Blackness in technology as manifested by underrepresentation among faces in image training sets. My goal is for white students to develop both empathy and understanding related to this form of oppression and to identify ways to work towards greater justice in computing.

As we enter this difficult conversation space, I conduct a short grounding in and give permission for students to move and breathe as needed, particularly students of color. I pay attention to potential power dynamics in the classroom by offering students to choose their own discussion groups instead of random assignment, which I do throughout the first half of the semester to help students meet others in the class. Since I have students writing weekly reflections, I also get to know which students have expertise, such as a major in Africana Studies or Gender and Sexuality Studies that may be important if I need to intervene in group formation in any way.

I then introduce Joy Buolamwini as a graduate student and researcher at the Massachusetts Institute of Technology who founded the Algorithmic Justice League, and preface the series of three videos as an evolution of her scholarship with respect to algorithmic justice. In successive order and with small individual reflection breaks, I show the following three video clips:

1. Ted Talk: The Coded Gaze (Boulamwini, 2021b)

2. Gendershades (Boulamwini, 2021c)

3. Ain't I a woman? (Boulamwini, 2021a)

I also emphasize that I appreciate the rich infusion of visual poetry aside the computer science justice issue of the last piece, and mention that it is why Joy Buolamwini is called the "Poet of Code." After another short individual reflective break, I again ask students to discuss in groups the "take-home messages" for these videos.

## 3. Reflections and Conclusions

In our discussions at the end of class, students never fail to mention the quote from the first video "Who codes matters, how we code matters and why we code matters (Boulamwini, 2021b)." Many semesters, I have had someone ask about fixing the particular algorithm or the particular training data set. In addition to acknowledging the fix for the specific problem at hand, I steer them to recognizing a systemic problem beyond a particular example. I have also had women, particularly Black women, so inspired by Joy Buolamwini that they have decided to pursue a flavor of computer science for their major. In addition to these anecdotal observations, we have documented steady gains in students' assessment of their own achievement of learning objectives throughout the semester (Taylor, 2021).

Despite the discussion here of a "module," the entire course is structured in such a way such that this discussion about marginalization and bias in computing and data science are not a one class event, but part of a broader quantitative critical inquiry arc in the class. The implementation of inclusive pedagogies such as the use of open source software and educational resources, co-construction of knowledge, and attention to small group power dynamics are meant to cultivate a supportive community learning journey that can foster this discourse.

Most of the attention on the benefits of Open Educational Resources are on the redistributive properties of social justice - shifting financial burdens on students away to gain full participation in the course experience (Lambert, 2018). Lambert points out two additional axes of social justice for the classroom, recognitive justice, the intentional inclusion of diverse voices and viewpoints, and representational justice, which allows marginalized people to speak for themselves. While Bergstrom and West are entertaining and easy for students to follow, their worldviews, to my knowledge, are that of white (cis)men. The series of Joy Buolamwini videos to help accomplish a course experience in which recognitive and representational justice are also present, where the Black woman is both computer scientist and social justice activist, as well as telling her story in her own words.

In preparation to facilitate such discussions in class, I point readers to scholars and scholarship in science and technology studies in addition to the resources above. Books such as *Race after Technology* (Benjamin, 2019), *Algorithms of Oppression* (Noble, 2018), and *Weapons of Math Destruction* (O'Neil, 2016) discuss these topics at a greater detail. I also recommend discussing class facilitation around topics relating to race, gender, and sexuality with colleagues who teach courses in the social sciences and/or one's office of equity and inclusion or intercultural education where staff are professional facilitators of such conversations on campus.

Students will not leave this course with the ability to perform machine learning techniques. However, I argue that creating the foundation to have critical discussions of such techniques will lead to deeper insight and better science when they reach courses which do teach these techniques. In addition, all corners of our data-driven economy are increasingly dependent on the results of such techniques. Therefore we have an obligation to be able to teach computational approaches such as machine learning to audiences beyond computer science and data science students and in our general education quantitative literacy courses.

The emphasis of this course is both literacy and critique and this particular module displays a heavier critique frame that explores racism and bias. However, we do spend time in the course discussing science's strive for objectivity through building a body of evidence and instituting structures such as peer review. We discuss the benefits of science and the evidence that science works (for example - planes fly and diseases like polio and measles have been nearly eradicated through vaccines). I argue that, regardless of the particular course context, making intentional space for the critical conversations is crucial to make sure that the world we create through our "objective" science is a world in which everyone can truly prosper. Ethical and equity-minded discussions should not be relegated to a stand alone ethics courses, but should be infused throughout the computer science curriculum at all levels.

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
