# OpenReview forum: "Teaching Machine Learning in the Context of Critical Quantitative Information Literacy"
_ecmlpkdd.org/ECMLPKDD/2021/Workshop/TeachML — TeachML 2021_

### Official Review · Reviewer_ZBEa · 2021-07-15
**an interesting description of a quantitative literacy course that is trying to do a lot of things at once**

**Rating:** 8
**Confidence:** 3

**Review:**

As a reviewer new to the workshop, I commend that author of the paper for being thorough in their presentation of their version of the Calling Bull course, which seems to be fairly well focused on calling bull on the ways that algorithms/ML can embed racist/sexist bias while seemingly being "objective." In framing my understanding in that fashion, I think I've highlighted what I think is a missing element in the course as presented here: a consideration of scientific objectivity, which is, to some degree built into the very notion of "calling bullshit." This is the fine line that a course dealing with the intersection of math and science with the social construction of reality is going to have to confront at some point. Embrace the line, I say. It's there. And to be clear, I think the course does some of this, but obliquely. If oblique is the pedagogical intent, then I think the author should make that clear and argue for it. (I can see for reasons for doing so, though in making bias so central to the course, it will have to be argued with care.)

A reader's ability to gauge the effectiveness of such a course is difficult at best. I do worry that by being so front-loaded on bias that perhaps the authors miss an opportunity to ensorcel students with all the "cool" things ML does and then making them realize what damage it does: it's rather like eating the tasty strawberries first and then realizing the terrible working conditions of migrant farm workers. Now you're implicated/imbricated. But such sequencing decisions are always best left to people actually having to do the work. As a reader, I wish there was room to have a fuller sense of the readings involved and, in particular, the assignments.

Finally, just as a note, there are a fair number of revision artifacts in the document, which occasionally had me reading sentences several times. I was curious, for example, about how on the first page one could "bridge multiple disciplines with computation and digital," if only because I couldn't help but read digital as an adjective. So, first, the parallelism was off to my mind and, second, I couldn't help but wonder "digital what?" There are a number of instances where such changes need to be fully realized.

---

### Official Review · Reviewer_BNsH · 2021-07-15
**A very interesting approach to look at ML in a course from the perspective of even negative consequences of using the technology**

**Rating:** 8
**Confidence:** 3

**Review:**

A very interesting approach to a course. The paper describes part of an information literacy course and extends the Calling Bull curriculum with programming instruction and projects in R, and supplemented with daily data visualization activities.

The authors present their instructional design in detail with many sources, and emphasize critical discussion of basic machine learning techniques rather than technical content of machine learning techniques as a learning objective.  Due to the good presentation, an adoption in own courses seems quickly and beneficially possible.

How far such a focus on the pure dangers and problems of using such techniques, does not lead to a general rejection of ML or technical solutions in general, would have to be discussed. A balanced presentation of risks and opportunities seems to make more sense. Nevertheless, the approach is extremely interesting and should definitely be discussed in the community.

A few suggestions on how we could approach the discussion:
* It remains unclear exactly how the module fits into the curriculum of the degree program. Perhaps this is clear from a US perspective, I found it difficult to compare this with European study modules.
* How do students respond to the course and content?
* Is there an evaluation planned that shows a gain in competency among students in the intended learning objectives?
* How do the lecturers of the technical modules react to this module, which is rather critical of ML?

---

### Decision · Program_Chairs · 2021-07-21

**Decision:**

Accept

**Comment:**

Congratulations! The reviewers agree that this paper should be accepted.

Camera-ready version is due August 18, 2021. As you prepare the camera ready version, please take the reviewers comments into consideration.

We look forward to your participation at the workshop on September 13, 2021. We invite you also to join us for the satellite event on September 08, 2021. Schedules for both the workshop and the satellite event will be forthcoming.